# Performance of Satellite-Based Evapotranspiration Models in Temperate Pastures of Southern Chile

**Italo Moletto-Lobos [1], Cristian Mattar [2,\*] and Jonathan Barichivich [3,4]**

1   Laboratory for Analysis of the Biosphere (LAB), University of Chile, Santiago 1030000, Chile; italo.moletto@um.uchile.cl
2   Laboratory of Geosciences, University of Aysén, Calle Obispo Vielmo N_62, Coyhaique 5952039, Chile
3   Instituto de Conservación Biodiversidad y Territorio, Universidad Austral de Chile, Valdivia 5090000, Chile; oreobulus@gmail.com
4   Laboratoire des Sciences du Climat et de l'Environnement, IPSL, CNRS/CEA/UVSQ, 91191 Gif sur Yvette, France
\*   Correspondence: cristian.mattar@uaysen.cl

**Abstract:** Farmers in the temperate zone of southern Chile have started to irrigate historically rainfed pastures during recent years to reduce dairy productivity losses against increasingly severe summer droughts. The lack of information on pasture water requirements (i.e., evapotranspiration), however, hampers the implementation of efficient irrigation programs. Here, we use in-situ observations to evaluate the skill of four remote sensing Surface Energy Balance (SEB) models and two satellite-based global evapotranspiration products (PML_V2 and GLEAM) to estimate actual evapotranspiration ($ET_a$) of pastures in southern Chile during 2014–2017. Daily $ET_a$ measured at an evaluation site over the period ranges between 1.2 mm and 6.2 mm day$^{-1}$ during the growing season (October–March), with an annual maximum of about 4.8 mm day$^{-1}$ in January and a minimum 0.6 mm day$^{-1}$ in June. Only the Simplified SEB (SEBS) model and its operational variant (SSEBop) and the PML_V2 global evapotranspiration product perform well, capturing 63–79% of the variance of in-situ evapotranspiration with an error between 0.75 mm day$^{-1}$ and 1.1 mm day$^{-1}$. The readily available PML_V2 product can be used as a convenient way to determine average water footprint of pastures and the two SEBs models can be implemented to monitor irrigation requirements in near-real time from field to regional scales. These results demonstrated a high potential of satellite observations for monitoring evapotranspiration and quantify the water footprint of pastures in southern Chile for a sustainable irrigation practice.

**Keywords:** evapotranspiration; GLEAM; PML_V2; pasture; grassland; remote sensing; Chile

## 1. Introduction

The mid-latitude westerly storm tracks in southern South America are shifting poleward with anthropogenic climate change [1], exposing the temperate region to progressively drier summer conditions that are prevalent at lower latitudes under a drier Mediterranean-like climate regime. This ongoing regional drying is part of a much larger scale drying trend across the southeastern Pacific and is expected to continue into the future [2]. The intensification of summer droughts [3–6], particularly during 2015 and 2016, has produced severe drops in pasture productivity (see Figure 1b) and large losses to the dairy industry. Water management programs for pastures have been implemented for the first time in the region to maintain productivity levels during the critical dry period of the austral summer. These traditionally rainfed pastures cover about 1.32 million hectares in the lowlands [7] and sustain major agricultural and dairy production in the country [8]. Accurate evapotranspiration

estimates and timely monitoring of pasture water requirements are necessary for the implementation of efficient irrigation programs, although this information is difficult to obtain at regional scale because of the scarcity of in-situ measurements for evapotranspiration retrievals over this large region.

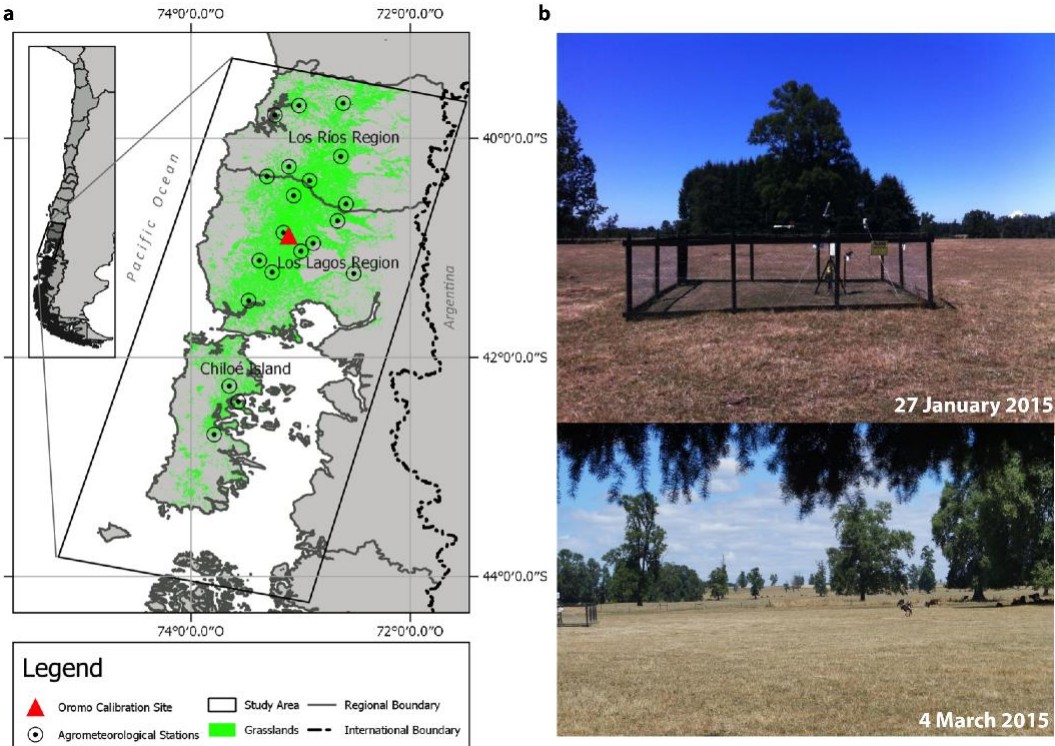

**Figure 1.** Study region and view of pastures at Oromo calibration site during the drought of the austral summer 2014–2015. (**a**) Temperate pastures in southern Chile (green shading), location of the agrometeorological monitoring stations (circles) and the Oromo calibration site (red triangle). (**b**) View of the monitoring station in Oromo and surrounding pastures during the middle and later part of the summer drought of 2014–2015.

Quantifying evapotranspiration is crucial for determining irrigation needs and the water footprint of pasture landscape systems [9–13]. A widely used approach to determine crop water consumption for irrigation management is through the estimation of $ET_a$ from the surface energy balance (SEB) based on high-resolution satellite observations of the land surface and meteorological data [14–17]. The SEB models can be categorized as one-source, two-layer, two-patch, dual-source, multi-patch, and multi-layer models. One-source models avoid the distinction in the aerodynamic resistance ($r_{ah}$) contribution between soil and vegetation [18]. The two-layer models consider the individual contribution of soil and vegetation to the total heat flux [19]. In contrast, the two-patch models estimate heat fluxes on the soil and vegetation component independently [20]. Hybrid models of dual source originate from the combination of the patch and layer approaches, resulting in a hybrid dual-source model [21]. The multi-patch models consider a greater spatial heterogeneity within each cell by dividing the surface into multiple patches, while the multi-layer models consider the vertical heterogeneity of surface conditions [22,23].

One-source models such as SEBAL (Surface Energy Balance Algorithm for Land), METRIC (Mapping EvapoTranspiration at high Resolution with Internalized Calibration), SEBS (Surface Energy Balance System) and SSEBop (Operational Simplified Surface Energy Balance) have been widely used to estimate $ET_a$. These models have shown good performance over terrestrial ecosystems and crops [24–27]. The $ET_a$ estimated by SEB models at basin or regional scales can be spatialized by remote sensing techniques, which allow to monitor different crops using currently available satellite data

from platforms such as the Copernicus Program, Earth Resources Observation System (EROS), and the recently launched ECOSTRESS [28,29]. These products provide a range of frequent and spatially continuous images of surface biophysical variables that influence $ET_a$ [30]. The resulting $ET_a$ retrievals can be directly validated using site-level in situ measurements of evapotranspiration from a range of field methods (e.g., Bowen ratio system, eddy covariance, scintillometry, etc.) to provide an accurate operational monitoring at larger regional scales for optimal water crop irrigation [31,32]

SEB models could provide an efficient approach to estimate and monitor water consumption of pastures in southern Chile to face the prospects of increasing water scarcity and uncertain sustainability of current pasture production systems. Here, we evaluate the performance of four SEB models (SEBS, SEBAL, METRIC and SSEBop) to estimate regional $ET_a$ from remote sensing observations. Their performance is compared to that of the readily available Global Land Evaporation Amsterdam Model (GLEAM 3.2b; [33,34] and Penman-Monteith-Leuning (PML_V2; [35–37] satellite-based global products. The structure of the paper is detailed as follows: Section 2 presents the data used and the study area, Section 3 describes the methodology, Section 4 presents the results and analysis, Section 5 presents the discussion and Conclusions.

## 2. Study Area and Datasets

### 2.1. Study Area

The study region includes all the temperate pastures in southern Chile, covering a surface area of 1,324,335 hectares between 39°20′ and 43°30′ S across Los Ríos and Los Lagos Districts (Figure 1a). This region includes the lowlands along the central depression of mainland Chile and the large island of Chiloé in the southwest. According to the Koppen–Geiger classification system, the regional climate corresponds to marine west coast (Cfb), with an average annual temperature of 10 °C and annual rainfall of 2100 mm [38]. The wet season extends from March to November but most precipitation falls in winter (44%) when the mid-latitude austral storm tracks move northward into the region. A dry season develops during the austral summer due to a southward expansion of the Mediterranean-like regional circulation system prevailing year-round to the north, with minimum seasonal soil moisture levels and frequent drought conditions occurring during February and early March (Figure 1b).

### 2.2. Satellite and Ancillary Data Used to Drive the Models

A total of 125 images of the Surface Reflectance product (Level 2-A) and the digital numbers (ND) of thermal bands were processed from Landsat 7 Enhanced Thematic Mapper Plus (ETM+) and Landsat 8 Operational Land Imager and Thermal Infrared Sensor (OLI/TIRS) [39,40] for the period 2014–2017 (available at https://earthexplorer.usgs.gov). All images were filtered by cloud cover using the C Function of Mask (CFMask) algorithm [41] and then resampled to the spatial resolution of the Landsat 8 thermal band (100 × 100 m).

Land surface emissivity ($\varepsilon_\lambda$) was obtained from the ASTER Global Emissivity Dataset (ASTER GED). This product has global coverage and includes the five thermal bands from the ASTER sensor between 2000 and 2008 [42,43]. The topography of the study area was derived from the SRTM (Shuttle Radar Topography Mission) Digital Elevation Model product [44]. Surface land cover types for the region were extracted from a land cover product based on Landsat 8 imagery at a spatial resolution of 30 m [7].

The Atmospheric Correction Parameter Calculator (ACPC) [45] was used in order to obtain atmospheric downward radiance ($L_d$), ascending radiance ($L_u$) and transmittance ($\tau$) for land surface temperature ($T_s$) retrievals over the study area. These parameters were obtained for the center latitude and longitude of each Path and Row of Landsat during the study period from July 2014 to December 2017. Earlier work demonstrated the capability of ACPC for $T_s$ retrievals over the study region in relation to MOD07 and AIRS datasets [46].

*2.3. Global Evapotranspiration Products*

The Global Land Evapotranspiration model from Amsterdam (GLEAM 3.3b) and Penman-Monteith-Leuning global $ET_a$ products were used to evaluate our implementation of the four-remote sensing SEB ET models. The GLEAM model estimates daily terrestrial evapotranspiration components and root-zone soil moisture from satellite data at 0.25° spatial resolution [33]. Potential evaporation ($ET_0$) is estimated using the Pristley–Taylor equation for the land fractions of bare soil, tall canopy and short canopy. Then, $ET_a$ is obtained by multiplying $ET_0$ with an evaporative stress factor based on microwave Vegetation Optical Depth (VOD) and satellite-based estimates of root-zone soil moisture. The Penman–Monteith–Leuning (PML_V2) product uses estimates of surface conductance ($G_s$) to describe the canopy-soil conductance to water flux at a moderate resolution of 500 m [37]. Actual evapotranspiration in PML_V2 relies mainly on the $G_s$ retrieval from remotely sensed land cover, Leaf Area Index (LAI) and Gross Primary Productivity (GPP).

*2.4. Ground Data*

Air temperature ($T_a$), Relative Humidity (RH) and Wind speed (u) measured at 2 m above the surface were obtained from 20 agrometeorological stations of the Chilean Agricultural Research Institute (INIA; available at: http://agromet.inia.cl/. Moreover, the automated weather station at Oromo Calibration Site (OCS) from LAB-Net [47] was used to validate surface temperature ($T_s$) and actual evapotranspiration ($ET_a$) estimates. Unlike the agrometeorological stations, this station also records radiative variables of the surface energy budget such as infrared temperature (TIR), Net Radiation ($R_n$) and soil heat flux (G). The OCS data can be freely obtained from http://biosfera.uchile.cl/ln_oromo.html.

## 3. Methods

*3.1. Surface Temperature ($T_s$) Estimation*

The $T_s$ was calculated for ETM+ and TIRS through the single-channel algorithm [48,49]

$$T_s = \gamma \left[ \frac{1}{\varepsilon_\lambda} (\psi_1 L_{sen} + \psi_2) + \psi_3 \right] + \delta \tag{1}$$

where $L_{sen}$ [W m$^{-2}$ sr$^{-1}$ μm$^{-1}$] is the radiance of the sensor in the thermal spectrum obtained by the radiometric calibration of the digital numbers of band 6 from ETM+ and band 10 from TIRS. $\varepsilon_\lambda$ was obtained from ASTER GED. The $\psi_1$, $\psi_2$ y $\psi_3$ variables are atmospheric functions, which depend on $\tau$, $L_d$ (W m$^{-2}$ sr$^{-1}$ μm$^{-1}$) and $L_u$ (W m$^{-2}$ sr$^{-1}$ μm$^{-1}$]). The $\gamma$ and $\delta$ are parameters that depend on $L_{sen}$ and the brightness temperature of sensor ($T_b$) (K). The validation of $T_s$ was carried out following the method proposed by Guillevic et al. [50].

*3.2. Net Radiation Estimates*

The SEB models used in this work rely on the spatialized net radiation flux ($R_n$) as a key variable for $ET_a$ retrievals because $R_n$ is partitioned into sensible heat flux (H), latent heat flux ($\lambda E$) and soil heat flux (G). $R_n$ was obtained using Equation (2):

$$R_n = (1 - \alpha)R_{swd} + \varepsilon_\lambda R_{lwd} - \varepsilon_\lambda \sigma T_s^4 \tag{2}$$

where $\alpha$ is the surface albedo, $R_{swd}$ (W m$^{-2}$) is the descending short-wave radiation, $R_{lwd}$ (W m$^{-2}$) is the long-wave descending radiation and $\sigma$ is the Boltzmann constant. The $\alpha$ calculation was done following the method proposed by [51]:

$$\alpha_{sw} = 0,356\alpha_b + 0,130\alpha_r + 0,373\alpha_{nir} + 0,085\alpha_{sw1} + 0,072\alpha_{sw2} - 0.0018 \tag{3}$$

where $\alpha_{sw}$ is the broadband surface albedo for the shortwave spectrum and $\alpha_b$, $\alpha_r$, $\alpha_{nir}$, $\alpha_{sw1}$ and $\alpha_{sw2}$ are the narrowband surface albedo values for the blue, red, near infrared and shortwave infrared spectrum, respectively. Although the albedo estimate requires the Bidirectional Reflectance Distribution Function (BRDF), it has been shown that for evapotranspiration retrievals bi-directional surface reflectance derived from satellite can be used for $ET_a$ maps [52]. $R_{swd}$ was computed following the method of [53], using extraterrestrial radiation ($R_A$) and Inverse Distance Weighted (IDW) interpolated near-surface air temperature ($T_a$), maximum daily temperature ($T_x$) and daily minimum temperature ($T_n$):

$$R_{swd} = R_A\left(A\left(1 - \exp(-B\Delta T)^C\right)\right) \tag{4}$$

The values of A, B and C were derived from the constant proposed by [53]. Then, $R_{lwd}$ was calculated through Stefan–Boltzmann equation. Atmospheric emissivity was estimated with the method proposed by [54] using $T_a$ and actual vapor pressure (e) as input.

$$R_{lwd} = 1.24\left(\frac{e}{T_a}\right)^{\frac{1}{7}} \sigma T_a{}^4 \tag{5}$$

Finally, $R_n$ was validated against ground measurements in the Oromo calibration site at the same time of the satellite overpass.

### 3.3. Surface Energy Balance Models and Eapotranspiration Modelling

The SEBS, SEBAL, METRIC and SSEBop models are all based on the surface energy balance (Equation (5)):

$$R_n = H + \lambda E + G_0 \tag{6}$$

where $R_n$ is net radiation [W m$^{-2}$], H is the sensible heat flux [W m$^{-2}$], $\lambda E$ to the latent heat flux [W m$^{-2}$] and $G_0$ is the soil heat flux [W m$^{-2}$], which is negligible for daily values [55].

The SEBS model uses meteorological and remote sensing data in order to estimate $ET_a$ and evaporative fraction ($\Lambda$), which is adjusted to the sensible heat flux by the Monin–Obukhov similarity [56] and energy balance under dry and wet limits.

$$\Lambda_r = 1 - \frac{H - H_{wet}}{H_{dry} - H_{wet}} \tag{7}$$

where $\Lambda_r$ is the relative evaporative fraction, $H_{wet}$ is sensible heat flux at the wet limit through a combination equation similar to the Penman–Monteith equation, and $H_{dry}$ is the sensible heat flux at the dry limit condition ($H_{dry} = R_n - G_0$). In this way, the latent heat flux was estimated as:

$$\lambda E = \Lambda_r \cdot \lambda E_{wet} \tag{8}$$

Daily $ET_a$ [mm day$^{-1}$] was estimated as:

$$ET_a = \frac{\lambda E}{\lambda} 86400 C_{di} \tag{9}$$

where $\lambda$ is the latent heat of vaporization and $C_{di}$ is the ratio of daily and instantaneous net radiation [57].

SEBAL is a physical model which estimates H by the parametrization of the vertical difference between the aerodynamic temperature and the air temperature close to the surface (dT), assuming a linear relationship between $T_s$ and dT.

$$H = \rho C_p \frac{dT}{r_{ah}} \tag{10}$$

To calibrate dT, it is necessary to select anchor pixels for each image under extreme conditions (dry and wet). The pixel of the dry limit is generally a bare agricultural field where H is maximum

($H_{hot} = R_n − G_0$). The cold pixel can be found in water bodies or well-irrigated fields where $H_{cold} = 0$. Therefore, $ET_a$ is estimated daily through Equation (8).

METRIC is a variant of the SEBAL model. The main difference lies in the "self-calibration" done in each image, since METRIC applies on the 'cold-pixel', the λE estimation based on a reference ET over a well-watered alfalfa field ($ET_r$) from hourly weather information. In this way just like SEBAL, H is calibrated by correcting the buoyancy and parameterizing dT based on the anchor pixels for extreme conditions.

Finally, SSEBop is a simplified surface energy balance model. It estimates $ET_a$ (Equation (10)) from $ET_0$, Λ and a coefficient k that scales the reference grass vegetation experiencing maximum ET by an aerodynamically rougher crop.

$$ET_a = \Lambda k ET_0 \tag{11}$$

where k = 1, assuming a reference grass cover. The evaporative fraction was computed (Equation (11)) through $T_s$ and reference land surface temperature under dry and wet conditions ($T_{hot}$, $T_{cold}$) based on $T_s$ and $T_x$.

$$\Lambda = \frac{T_{hot} − T_s}{dT} = \frac{T_{hot} − T_s}{T_{hot} − T_{cold}} \tag{12}$$

dT was estimated using a reference value of $r_{ah}$ and maximum H for bare soil surface, where $r_{ah} = 110$ [s m$^{-1}$] [58] and $H = R_n − G_0$.

$$dT = \frac{110 R_n}{\rho_a C_p} \tag{13}$$

The SEBS, SEBAL, METRIC and SSEBop models were evaluated by comparing the estimated pixels over the Oromo calibration site with in situ pasture evapotranspiration ($ET_c$).

$$ET_c = k_c ET_0 \tag{14}$$

where $k_c$ is the crop coefficient, obtained from a linear function from Normalized Difference Vegetation Index (NDVI) from Landsat data over Oromo site. The $ET_0$ was estimated using the standardized Penman–Monteith equation [59]:

$$ET_0 = \frac{0.408\Delta(R_N − G_0) + \gamma\frac{900}{T_a} + u_2(e_s − e_a)}{\Delta + \gamma(1 + 0.34u)} \tag{15}$$

where u is wind speed [m s$^{-1}$], $e_s$ is the saturation vapor pressure (kPa), $e_a$ to the actual vapor pressure (kPa), and Δ is the slope of the vapor pressure [kPa °C$^{-1}$]. Modeled $ET_a$ was compared to measured $ET_c$ at the Oromo calibration site using the coefficient of determination ($R^2$), root mean square error (RMSE or RMSD), the standard deviation of the residual values (σ), the bias of residual values and the mean absolute error (MAE) statistics.

## 4. Results

### 4.1. Estimated Surface and Meteorological Fields

The spatial patterns of estimated $T_s$ for selected spring and summer days over the study domain are shown in Figure 2 along with a pixel-level validation against ground observations at the Oromo calibration site. The latitudinal temperature gradient is weaker than variations associated with vegetation cover and elevation. There is a clear contrast between warmer pastures in the interior lowlands and colder rainforests in the Andean and Coastal mountains to the eastern and western parts of the domain, respectively.

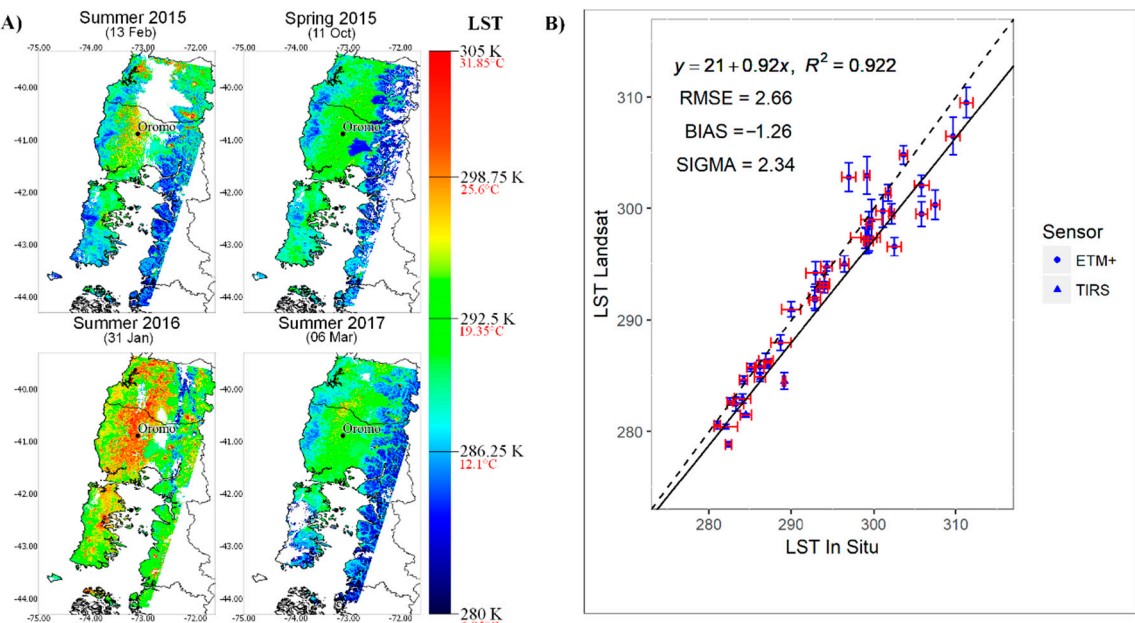

**Figure 2.** Estimates of surface temperature and comparison with in-situ measurements. (**a**) Satellite-based estimates of $T_s$ for the study area during selected spring and summer days. White gaps in the images represent regions with cloud cover. (**b**) Local comparison with in-situ measurements at the Oromo calibration site.

The region experienced a pronounced warming during 2016, with most of the pastures exceeding 25 °C (Figure 2a). The estimated high temperatures (>300 K) for this period compare well with in-situ measurements at the Oromo calibration site (Figure 2b). Atmospheric moisture demand increased rapidly, and pasture productivity declined (Figure 1b), leading to many farmers to irrigate these historically rainfed systems to maintain productivity levels. Satellite-based temperature estimates show a good linear fit with in-situ measurements ($r^2 = 0.92$) but have a cold bias of −1.26 K (Figure 2b).

Cloud cover can significantly reduce data availability during the wet and even the dry summer season, resulting in substantial spatial gaps in satellite-based radiation and surface temperature fields (e.g., summer 2015 in Figure 3). The interpolated meteorological fields of air temperature, instantaneous solar radiation and relative humidity capture the large-scale patterns of variation reasonably well but show substantial uncertainties at the pixel level due to the sparseness of the underlying network of agrometeorological stations (Supplementary Materials Figure S1). Pixel-level validation at our Oromo calibration site shows RMSE values of 6.12 [%] for relative humidity, 248 [W m$^{-2}$] for global radiation, 0.90 [K] for air temperature and 1.15 [m/s$^{-1}$] for wind speed (Supplementary Materials Figure S2).

Instantaneous net radiation for summer days averages 575.02 ± 91.84 [W m$^{-2}$]. As expected, $R_n$ was homogeneous over grasslands during clear sky-days such as 11 Oct 2015 and 31 January 2016 (Figure 3a). Under cloudy conditions, such as in 13 February 2015 and 6 March 2017, $R_n$ values are lower (~ 300 [W m$^{-2}$]) and more spatially variable. At the Oromo calibration site, the linear fit of satellite-based $R_n$ is 0.70 with an RMSE of 208.38 [W m$^{-2}$] and a substantial overestimation (BIAS of 175.25 [W m$^{-2}$]) due to the retrieved surface albedo for grasslands (Figure 3b). Indeed, the estimated surface albedo ($\alpha = 0.16$) is 9% lower than the average albedo measured in Oromo ($\alpha = 0.27$). In addition, the presence of clouds, as indicated by the high variability of in situ measurements, is associated with greater departures between satellite estimates and in-situ measurements.

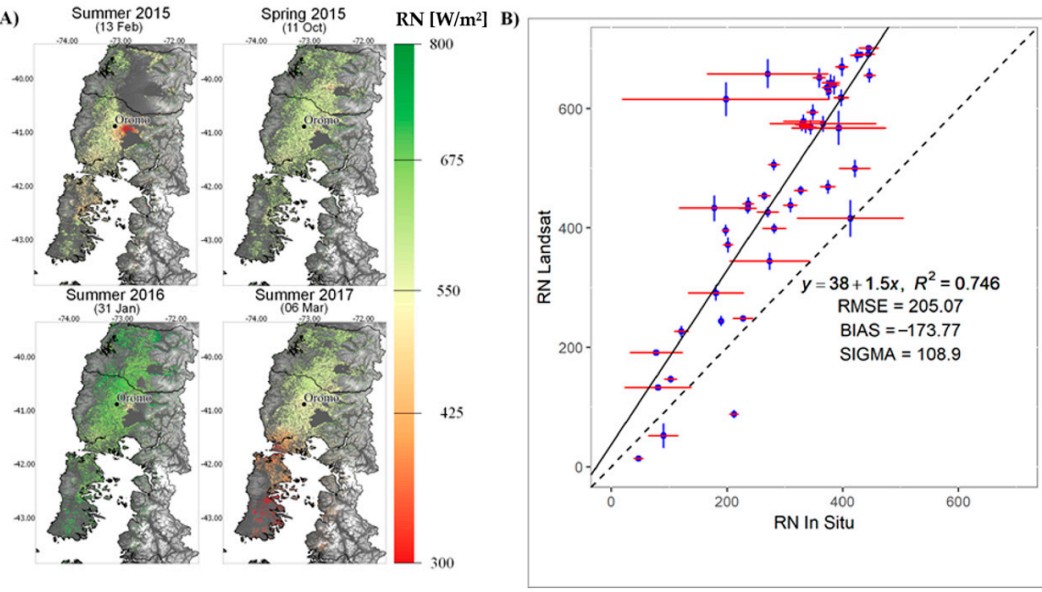

**Figure 3.** Estimates of net radiation and comparison with local measurements. (**a**) Satellite-based estimates of instantaneous net radiation [W m$^{-2}$] in pastures during selected spring and summer days. Note that the images correspond to the same dates than those shown in Figure 3 for surface temperature. (**b**) Local comparison against in-situ measurements at the Oromo calibration site.

### 4.2. In-situ and Modeled Grassland Evapotranspiration

Annual in situ precipitation and evapotranspiration in Oromo averaged 1178 and 517 mm over the study period, respectively (Table 1). This results in a positive net water flux into the surface on a yearly basis (661 [mm year$^{-1}$]) as annual evapotranspiration amounts to 44% of annual precipitation. However, during dry summers with precipitation under 100 [mm] (e.g., 2015 and 2016), evapotranspiration can exceed precipitation for up to 100 and 200%. For instance, during the extremely dry summer of 2015 rainfall reached a historical low of only 46.7 [mm] and soil moisture fell to 14–15%, but evapotranspiration was maintained at near average levels and reached 148.2 [mm] (Table 1). Seasonally, evapotranspiration reaches a maximum during spring (216.4 [mm]) when soils are wet and radiation is near its seasonal peak, and then it decreases towards the summer (166.8 [mm]), autumn (60.6 [mm]) and winter (48.2 [mm]) as water and then energy become limiting. The evapotranspiration estimates of the PML_V2 global product also follow the same pattern than the observations in Oromo and provide a longer record (Table 1).

The SEBS and SSEBop models produce similar spatial patterns of grassland evapotranspiration during summer days (Figure 4). In contrast, the SEBAL and METRIC models simulate substantially higher spatial variability in evapotranspiration. During the severe summer drought that affected the region in 2014–2015, soil moisture levels in the Oromo station by 13 February 2015 reached some of the lowest levels (0.15 [m$^3$ m$^{-3}$]) on the 2014–2017 record and surface temperatures exceeded 295 [K] (Figure 5b–c). During this hot summer drought day, the SEBS and SSEBop models estimated an evapotranspiration flux of 4.38 ± 1.0 and 4.03 ± 0.9 [mm day$^{-1}$], respectively (Figure 4). For the same day, the SEBAL and METRIC models estimated an evapotranspiration flux of 3.89 ± 1.75 and 4.24 ± 2.49 [mm day$^{-1}$], respectively. The estimated evapotranspiration flux under drought conditions is about 30% lower than the estimates for a similar date (31 January) during the following summer in 2016, when SEBS and SSEBop suggest an evapotranspiration flux of 5.59 ± 0.86 and 5.79 ± 0.87 [mm day$^{-1}$], respectively.

Figure 5 shows a comparison between time series of evapotranspiration obtained from the SEB models and estimates from global ET products and in situ estimates at the Oromo calibration site. Persistent cloud cover precluded evaluating the performance of the models during winter. The

estimates from the SSEBop model closely follow in-situ evapotranspiration, particularly during autumn and summer (Figure 5a). The rest of the models strongly overestimate evapotranspiration during summer. The magnitude of the overestimation in SEBAL, METRIC and SEBS models exceeds 100% in January and February. However, the SEBS model matches SSEBop during autumn.

**Table 1.** Seasonal evapotranspiration (mm), precipitation (mm) and soil moisture (%) in Oromo over the period 2014–2017. In-situ evapotranspiration (bold) and the corresponding estimate from the PML_V2 global product (brackets) are given for annual and austral spring (September–November), summer (December–February), autumn (March–May) and winter (June–August) seasons.

|  | **2014** | **2015** | **2016** | **2017** | **Mean** |
|---|---|---|---|---|---|
| Evapotranspiration (mm) |  |  |  |  |  |
| Spring | [222.8] | [227.8] | **238.5** [239.5] | **194.3** [242.8] | **216.4** [233.3] |
| Summer | [176.8] | **148.2** [170.8] | **189.2** [226.1] | **163.1** [254.9] | **166.8** [204.9] |
| Autumn | [71.2] | **59.7** [63.8] | **70.4** [75.9] | **61.7** [79.0] | **60.6** [69.4] |
| Winter | [71.9] | **27.7** [65.9] | **64.4** [71.4] | **52.6** [68.2] | **48.2** [72.5] |
| Annual | [542.8] | [528.4] | **562.2** [612.9] | **471.8** [635.9] | **517.0** [580.0] |
| Precipitation (mm) |  |  |  |  |  |
| Spring | 130.8 | 150.6 | 231.6 | 228.8 | 185.5 |
| Summer | - | 46.7 | 81.7 | 222.8 | 117.1 |
| Autumn | - | 442.2 | 183.2 | 537.1 | 387.6 |
| Winter | 490.7 | 543.8 | 371.9 | 509.9 | 487.9 |
| Annual | 621.5 | 1183.3 | 868.4 | 1498.6 | 1178.0 |
| Soil moisture 7 cm (%) |  |  |  |  |  |
| Spring | 32.2 | 29.8 | 31.1 | 32.4 | 31.3 |
| Summer | - | 14.3 | 13.9 | 22.6 | 16.9 |
| Autumn | - | 35.9 | 30.5 | 36.1 | 34.2 |
| Winter | 45.9 | 42.2 | 41.3 | 41.9 | 43.0 |
| Soil moisture 20 cm (%) |  |  |  |  |  |
| Spring | 30.7 | 28.2 | 29.8 | 30.2 | 29.3 |
| Summer | - | 15.7 | 16.3 | 24.3 | 18.8 |
| Autumn | - | 28.6 | 27.9 | 34.1 | 30.3 |
| Winter | 38.2 | 35.6 | 35.5 | 37.1 | 36.6 |

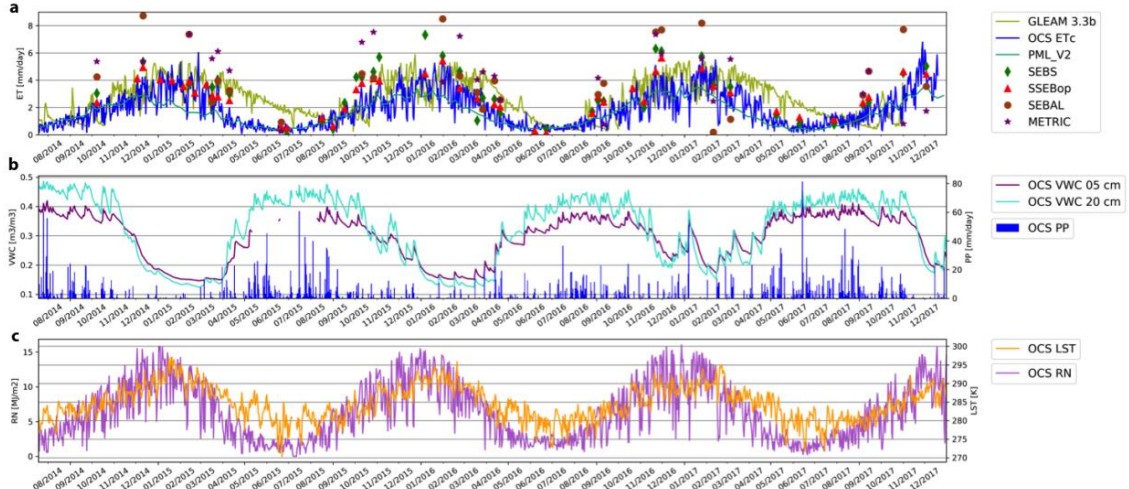

**Figure 4.** Comparison of evapotranspiration maps for pastures during summer days produced by each model (SEBS, SSEBop, SEBAL, METRIC). (**a**) 13 February 2015. (**b**) 31 January 2016. (**c**) 6 March 2017.

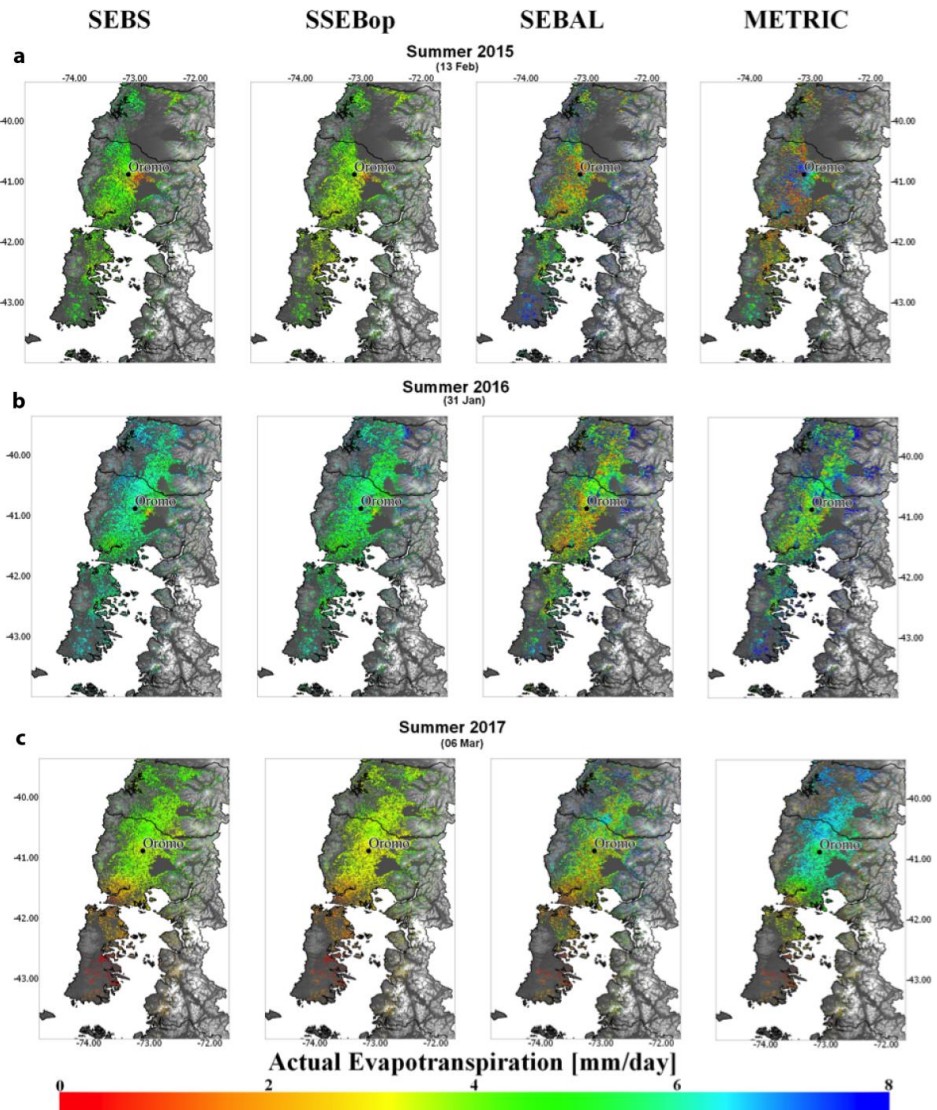

**Figure 5.** Evapotranspiration estimates and daily surface meteorology at the Oromo calibration site (OCS) between August 2014 and December 2017. (**a**) Comparison of satellite-based and measured daily evapotranspiration. Measured $ET_c$ in Oromo (blue line) is compared with estimates by SEB models (points) and GLEAM 3.3b (yellow line), and PML_V2 (turquoise line) products for the nearest pixel to the station. (**b**) Daily rainfall (blue bars) and soil moisture (purple, turquoise lines) in Oromo. (**c**) Net daily radiation (purple) and surface temperature (orange) measured in Oromo.

The performance of the global evapotranspiration product GLEAM 3.3b at the Oromo site contrasts strongly during the spring-summer and autumn-winter periods (Figure 5a). The increase in local evapotranspiration with rising temperature and radiation during the spring and early summer is well captured, but evapotranspiration during late summer and autumn is systematically overestimated by around 2 [mm day$^{-1}$]. The poor performance in autumn is likely related to a mixing of land cover types and misrepresentation of local vegetation phenology in this product due to its coarse resolution (~25 km). The higher-resolution PML_V2 product has a better performance than GLEAM 3.3b and follows more closely in-situ evapotranspiration through the year, although there is some systematic underestimation (<0.5 [mm day$^{-1}$]) during summer (Figure 5a). The PML algorithm is likely overestimating moisture limitation to evapotranspiration during summer, when surface soil moisture typically falls below 20% (Figure 5b).

Large differences between SEBAL and METRIC models during spring and autumn were observed in Oromo when soil moisture peaked following rainfall events, causing an underestimation of H at the anchoring pixels and thus increasing the uncertainty of estimated $ET_a$. During summer 2017, a strong rainfall event increased soil water availability and increased $ET_a$ at Oromo from 2 to 5 [mm day$^{-1}$] (Figure 5a,b). In addition, this event was also revealed by SSEBop, which matches in-situ estimates and other models such as SEBS and METRIC. The rest of the models did not show the impact of rainfall during summer under drought conditions.

The relative performance of SEB and global models against Oromo measurements is shown in Figure 6. SSEBop and SEB are by far the best models in terms of error and correlation with observations, followed by the global evapotranspiration product PML_V2 (Figure 6a). These products are able to explain between 63 and 79% of the variance ($R^2$) of the in-situ observations (Figure 6b). The SEBS model accounted for 74% of the variance in observed evapotranspiration ($ET_a$), with relatively low error (RMSE 1.08 [mm day$^{-1}$] and MAE of 0.85 [mm day$^{-1}$]; Figure 6b) but slight underestimation when estimated instantaneous $R_n$ is lower than 300 [W m$^{-2}$] (bias: 0.41 [mm day$^{-1}$]). SSEBop performed slightly better than SEBS ($R^2$: 0.79, RMSE: 0.68 mm day$^{-1}$, MAE: 0.55 [mm day$^{-1}$]) with the lowest bias (0.09 [mm day$^{-1}$]). In contrast, SEBAL presents a low statistical fit ($R^2$: 0.24) in addition to high error, with a RMSE of 2.46 [mm day$^{-1}$] and MAE: 2.01 [mm day$^{-1}$] mainly due to its high bias (1.22 [mm day$^{-1}$]) during summer. METRIC had the poorest performance, with maximum RMSE of 2.57 [mm day$^{-1}$], high dispersion of estimated values ($\sigma$: 2.11 [mm day$^{-1}$]) and close to null correlation with Oromo measurements ($R^2$: 0.05). Therefore, the values estimated by METRIC are not representative over the study area. GLEAM shows an overall overestimation, with a bias up to 0.75 [mm day$^{-1}$] and consequently low $R^2$ (0.35) and high RMSE 1.45 [mm day$^{-1}$]). In contrast, PML_V2 shows a better agreement with in situ observations ($R^2$: 0.63, RMSE: 0.88 [mm day$^{-1}$]) than GLEAM but it is the only model that underestimates (bias $-0.28$ [mm day$^{-1}$]) high evapotranspiration values (Figure 6b).

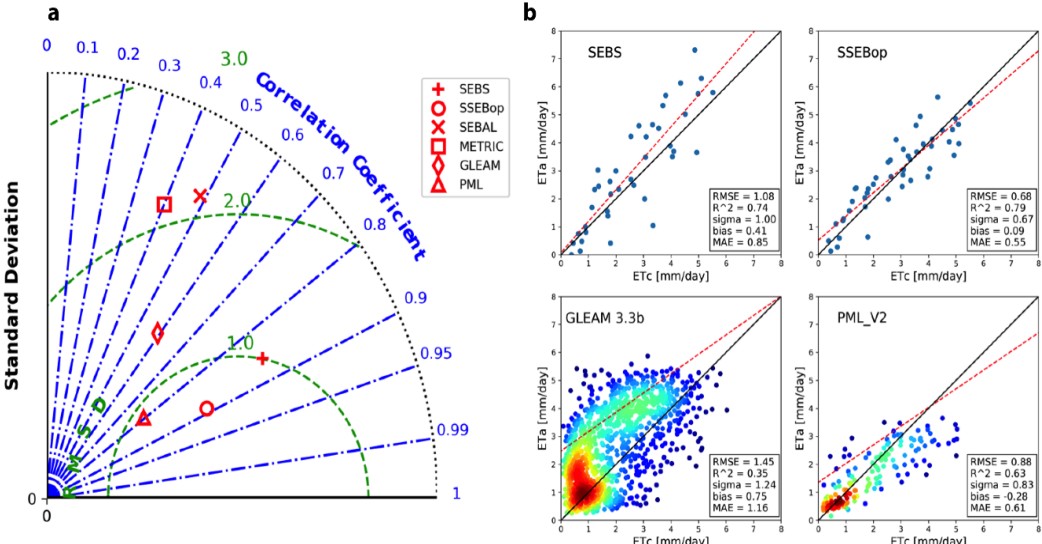

**Figure 6.** Performance of SEB models and global evapotranspiration products in Oromo. (**a**) Taylor diagram of evapotranspiration models in relation to the in-situ observations. The dashed green line denotes the root mean standard deviation (RMSD), while the dashed blue line indicates the correlation coefficient with the observations. (**b**) Scatterplots comparing estimated ($ET_a$) and measured ($ET_c$) evapotranspiration in Oromo. The worst performing models SEBAL and METRIC are not shown. The dashed red line represents the identity function, and the black line denoted the slope of the linear regression of each model. GLEAM and PML_V2 comparisons are shown as density scatterplots, with red and blue colors denoting high- and low-density areas, respectively.

## 5. Discussion and Conclusions

Summer evapotranspiration is becoming a crucial indicator for irrigation plans and drought management in the Chilean temperate pastures. A suite of four remote sensing surface energy balance (SEB) models and two readily available global evapotranspiration products were validated using ground observations. Surface temperature ($T_s$) was validated in a single point in the study area and the results are satisfactory, with an RMSE of 2.66 [K] for 42 days with data availability between 2014–2017. However, this validation could be improved by new measurement campaigns and equipping additional stations to measure $T_s$ and determine in-situ emissivity, considering the phenology of grasslands under moisture limitation. Variations in phenology affect the emissivity by up to 3% due to surface moisture [60,61] and therefore impact the proportion of soil and vegetation cover. Our results provided the first validation of surface temperature in southern Chile at high spatial resolution to estimate accurately crop water demand at regional scale.

The models SSEBop and SEBS do not solve the dT under the selection of anchors pixels and probably the assumption of H and $R_n$ can be more adaptable for retrievals at smaller spatial scales. The models SEBAL and METRIC require the selection of an extreme condition to estimate H and λE. This selection procedure affects the performance of those models over temperate pastures in the southern Chile. Previous studies have shown that the assumption of extreme conditions lead to uncertainties in $ET_a$ due to the subjectivity of pixel selection and range of soil moisture conditions in the study domain [62–64]. However, if the application of these models in the study region is limited to smaller areas, their performance might improve substantially. SEBS evapotranspiration estimates are well correlated with the observations in Oromo ($R^2 = 0.83$) but the high sensitivity of the model to $R_n$ results in an overestimation.

The global evapotranspiration products showed marked differences in their performance. GLEAM does not represent well $ET_a$ at the local scale (RMSE over 1.45 [mm day$^{-1}$]) and overestimates it during autumn and winter, likely because of mixed land covers in the coarse pixel of 25 km. The PML_V2 model, which is based on moderate-resolution biophysical estimation of $ET_a$ coupled with GPP, showed high accuracy over southern grasslands, with a correlation coefficient of 80% and RMSE lower than 1 mm day$^{-1}$. A spatial comparison between SSEBop and the global products resampled to the same resolution (Supplementary Materials Figure S3) showed that GLEAM does not have a good performance, reaching on average a $R^2$ of 0.4. In contrast, PML_V2 had an overall $R^2 = 0.8$ at exception of Chiloé Island, where the spatial correlation is lower (R = 0.3 to 0.5). This finding indicates that PML_V2 can be used as a high-quality reference for evapotranspiration at the scale of 500 m over the vast continental grasslands across Los Ríos and Los Lagos regions.

Annual precipitation is twice annual evapotranspiration in the region (Table 1), but summer evapotranspiration exceeded precipitation during recent droughts. For this reason, validating actual evapotranspiration at regional scale for grasslands of southern Chile using satellite and meteorological data is essential to accurately determine the actual water consumption of grasslands. Using this information, it is possible to promote efficient irrigation programs that can correct the water deficit during the critical spring-summer period. This can effectively reduce the water footprint of regional pastoral systems, avoiding the extra irrigation of grasslands and promoting better practices to reduce energy and water consumption. In addition, these data would facilitate the implementation of $ET_a$ monitoring aimed at reducing risk of water stress during the increasingly recurrent summer drought periods.

Taken together, our combined analysis and validation of the four remote sensing evapotranspiration models and the two global evapotranspiration products showed that the SEBS and SSEBop models can provide an accurate ($R^2$: 74–79 and RMSE < 1 [mm day$^{-1}$]) field-scale estimate of grassland evapotranspiration in the region, which can be complemented by readily available weekly estimates at 500 m from the global PML_V2 product. This finding is encouraging for the implementation of a near-real time monitoring tool for evapotranspiration in the region for the timely application of effective irrigation programs during the recurrent summer droughts. However, widespread irrigation

might increase the pressure on already limited water resources, and thus the practice has to be properly regulated and considered within a framework of sustainable agriculture and adaptation to climate change.

**Supplementary Materials:** The following are available online at http://www.mdpi.com/2073-4441/12/12/3587/s1: Figure S1. Meteorological data for the study area. (A) is the wind speed [m s$^{-1}$], (B) air temperature [K], (C) instantaneous global radiation [W m$^{-2}$] y (D) relative humidity [%]. The meteorological information is on a topography layer of the región represented in gray. Figure S2. Validation of relative humidity (RH), global radiation (Rx), air temperature (Ta) and wind speed (u). (A). Scatterplot of RH, Rx, Ta and u in Oromo Calibration Site (x-axis) and spatially interpolated variables (y-axis). (B) Evaluation of bias, sigma and RMSE at pixel scale over OCS. Figure S3. Spatial correlation between global models (GLEAM and PML) and SSEBop model.

**Author Contributions:** I.M.-L. and C.M. conceived and designed the research. I.M.-L., C.M. and J.B. performed the data processing and analysis. All authors contributed to the manuscript write-up and editing. All authors have read and agreed to the published version of the manuscript

**Funding:** This work was funded by the PYT-2016-0265 program of the "Fundación para la Innovación Agraria (FIA)" of the Chilean Ministry of Agriculture and the Regional Government of the Los Lagos Region, Chile.

**Acknowledgments:** The authors also thank the USGS, LP DAAC for provide ASTER GED, MODIS and Landsat Data. JB acknowledges support from the FONDECYT grant number 1171496. Finally, the authors acknowledge the useful support of Patricia Soto for editing the manuscript.

**Conflicts of Interest:** The authors declare no conflict of interest.

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
