# Peer review of "Performance of Satellite-Based Evapotranspiration Models in Temperate Pastures of Southern Chile"

_water, doi:10.3390/w12123587_

Round 1

Reviewer 1 Report

The article deals with interesting topic of performance of satellite-based evapotranspiration models in temperate pastures of southern Chile.
I have no objections in the whole manuscript. The concept is well designed, an overview of previous research is described and the methods used are well described.
My only complaint is related to the visual display of equations and figures, which in my opinion are not written in some equation editor (Latex or some other) so it should be corrected, and through the whole handwriting minus in units is not in the exponent.
Given such comments, I suggest accepting the manuscript with a minor revision.

Author Response

R.: We appreciate the evaluation. The revised the manuscript following the suggested points.

Reviewer 2 Report

The article concerns problems related to contemporary climate change, namely the evaluation of the performance of 4 SEB 83 models to estimate regional actual evapotranspiration (Eta) from remote sensing observations. The research is certainly not easy, and the obtained results are subject to high uncertainty. The article has a logical structure. The value of the research results obtained is purely theoretical due to large estimation errors.

I am rather critical of this type of research due to its uncertainty. However, I do not take away the authors' right to run them.

Comments

Is there only a problem of drought in the area under consideration or is there a problem of excess rainfall? In other parts of the temperate zone (e.g. northern), both phenomena are affected, which means an increase in climate variability. If droughts are to be more frequent, it means that there is a decrease in rainfall, or the temperature increase is so large that it contributes to the intensification of drought. I suggest a short discussion in this regard at the beginning of the article, supported by research results.

Other

 A bit too long abstract with too many redundant details. The abstract is intended to attract the reader to the article, not necessarily a summary of it. So I suggest that the Authors think about it. 1 sentence of the abstract is unclear: “Altered atmospheric circulation patterns due to anthropogenic climate change are intensifying the summer dry season in southern South America”. There are climate changes and there can be many reasons for this change. Anthropogenic climate change ?? It suggests that they can be quantified.

 Check if the formulas are written correctly, e.g. page 5 line 166 should be Rlwd (with subscript). Page 7, line 237: warming during the summer of 2016 ,? What does one year warming mean? Correct the title of Fig. 2: local measurements?

Table 1. Summary of seasonal evapotranspiration…. Title for improvement. This is not a summary? Units (mm?) are missing.

Author Response

Comments

Is there only a problem of drought in the area under consideration or is there a problem of excess rainfall? In other parts of the temperate zone (e.g. northern), both phenomena are affected, which means an increase in climate variability. If droughts are to be more frequent, it means that there is a decrease in rainfall, or the temperature increase is so large that it contributes to the intensification of drought. I suggest a short discussion in this regard at the beginning of the article, supported by research results.

R.: We have included a new paragraph in the discussion addressing the rainfall decrease and agriculture sustainability system.

Other

 A bit too long abstract with too many redundant details. The abstract is intended to attract the reader to the article, not necessarily a summary of it. So I suggest that the Authors think about it. 1 sentence of the abstract is unclear: “Altered atmospheric circulation patterns due to anthropogenic climate change are intensifying the summer dry season in southern South America”. There are climate changes and there can be many reasons for this change. Anthropogenic climate change ?? It suggests that they can be quantified.

R.: The abstract was revised.

 Check if the formulas are written correctly, e.g. page 5 line 166 should be Rlwd (with subscript). Page 7, line 237: warming during the summer of 2016 ,? What does one year warming mean? Correct the title of Fig. 2: local measurements?

R.: Several typos and formatting details were corrected.

Table 1. Summary of seasonal evapotranspiration…. Title for improvement. This is not a summary? Units (mm?) are missing.

R.: Title and units were improved.

Round 2

Reviewer 2 Report

The authors responded to the comments sufficiently.